Breast fibroblasts in both cancer and normal tissues induce phenotypic transformation of breast cancer stem cells: a preliminary study

http://orcid.org/0000-0001-6961-3532 Wang Bixiao 1
Xi Chunfang 1 2
Liu Mingwei 3
Sun Haichen 4
Liu Shuang 4
Song Lei 3
Kang Hua 1 kanghuamd@163.com
1 Department of General Surgery, Xuanwu Hospital, Capital Medical University , Beijing , China
2 Department of Breast Surgery, Shanxi Provincial People’s Hospital , Taiyuan, Shanxi , China
3 State Key Laboratory of Proteomics, National Center for Protein Sciences, Beijing Proteome Research Center , Beijing , China
4 Surgery Lab, Xuanwu Hospital, Capital Medical University , Beijing , China
Costa-Lotufo Leticia
Electronic publication date: 2018 May 15
Publication date: 2018
Volume: 6
Electronic Location ID: e4805
Received 2017 Dec 15; Accepted 2018 Apr 29
Copyright: © 2018 Wang et al.
Copyright year: 2018
Copyright holder: Wang et al.
License: This is an open access article distributed under the terms of the Creative Commons Attribution License, which permits unrestricted use, distribution, reproduction and adaptation in any medium and for any purpose provided that it is properly attributed. For attribution, the original author(s), title, publication source (PeerJ) and either DOI or URL of the article must be cited.
License URL: https://creativecommons.org/licenses/by/4.0/

Keywords: Cancer stem cell, Fibroblast, Breast, Phenotypic transformation, Breast cancer, ALDH1, CD44CD24

Funding: National Natural Science Foundation of China 81172517 Beijing Breast Prevention Institute of Breast Cancer Prevention and Treatment Research Fund This work was supported by the National Natural Science Foundation of China (No. 81172517) and Beijing Breast Prevention Institute of Breast Cancer Prevention and Treatment Research Fund. The funders had no role in study design, data collection and analysis, decision to publish, or preparation of the manuscript.

==============================
Background

Breast cancer stem cells (BCSCs) are associated with the invasion of breast cancer. In recent years, studies have demonstrated different phenotypes among BCSCs. Furthermore, BCSCs of diverse phenotypes are present at different tumour sites and different histological stages. Fibroblasts are involved in the phenotypic transformation of BCSCs. Cancer-associated fibroblasts (CAFs) participate in the induction of epithelial–mesenchymal transition, thereby promoting the acquisition of stem cell characteristics, but little is known about the role of normal fibroblasts (NFs) in the phenotypic transformation of BCSCs or about the effect of CAFs and NFs on BCSC phenotypes.

Methods

A total of six pairs of primary CAFs and NFs were isolated from surgical samples of breast cancer patients and subjected to morphological, immunohistochemical, cell invasion and proteomics analyses. After establishing a cell culture system with conditioned medium from CAFs and NFs, we used the mammosphere formation assay to explore the effect of CAFs and NFs on the self-renewal ability of BCSCs. The effect of CAFs and NFs on the phenotypic differentiation of BCSCs was further analysed by flow cytometry and immunofluorescence.

Results

The isolated CAFs and NFs did not show significant differences in cell morphology or alpha-smooth muscle actin (α-SMA) expression, but cell invasion and proteomics analyses demonstrated heterogeneity among these fibroblasts. Both CAFs and NFs could promote the generation of BCSCs, but CAFs displayed a greater ability than NFs in promoting mammosphere formation. Conditioned medium from CAFs increased the proportion of aldehyde dehydrogenase-1 positive (ALDH1+) BCSCs, but conditioned medium from NFs was more likely to promote the generation of CD44+CD24− BCSCs from MCF-7 cells.

Discussion

This study validated the heterogeneity among CAFs and NFs and expanded on the conclusion that fibroblasts promote the generation of cancer stem cells. Our results particularly emphasized the effect of NFs on the phenotypic transformation of BCSCs. In addition, this study further highlighted the roles of CAFs and NFs in the induction of different phenotypes in BCSCs.

Introduction

Cancer stem cells are a subpopulation of cancer cells that are characterized by self-renewal and multidirectional differentiation capabilities. Since breast cancer stem cells (BCSCs) were isolated in 2003 and shown to exhibit strong tumourigenicity (Al-Hajj et al., 2003), a large number of studies have described the use of CD44, CD24 and aldehyde dehydrogenase-1 (ALDH1) for the identification of BCSCs (Chiotaki, Polioudaki & Theodoropoulos, 2016; Yang et al., 2017). These studies have shown that BCSCs are associated with tumour progression such as recurrence, metastasis and treatment resistance, and consequently contributed to the poor outcome of breast cancer. However, not all BCSC markers are expressed in all breast cancer subtypes, and these markers may be limited to particular subtypes or associated with invasive subtypes of breast cancer (Boesch et al., 2016). For example, CD44+CD24− BCSCs are abundant in triple-negative breast cancer (Bernardi et al., 2012), and ALDH1+ BCSCs are associated with the BRCA1 mutational status (Somasundaram et al., 2016). In addition, these cell populations are enriched in primary tumour lesions after radiotherapy and chemotherapy but are seldom associated with either disease-free or overall survival (Li et al., 2008). However, ALDH1+ BCSCs are associated with HER-2 overexpression and loss of expression of oestrogen receptors and progesterone receptors (Ginestier et al., 2007), and the presence of this population may be used as an independent predictor of breast cancer prognosis (Mansour & Atwa, 2015). These findings indicate that BCSCs may vary by cancer subtype. Even for a particular breast cancer lesion, there may be differences in stem cell phenotypes at different tumour sites and different histological stages (Da Cruz Paula & Lopes, 2017). Therefore, we hypothesize that a process of transformation is involved in the variation of BCSC phenotypes.

The tumour matrix or the tumour microenvironment can affect the biological behaviour of cancer cells and plays an important role in the disease progression (Bussard et al., 2016). Fibroblasts are the major cellular components in the tumour microenvironment, and fibroblasts that are activated by cancer cells are called cancer-associated fibroblasts (CAFs). Many studies have shown that CAFs play an important role in tumour formation, development, invasion, metastasis and resistance to treatment by secreting exosomes or a variety of cytokines and factors that remodel the extracellular matrix (Buchsbaum & Oh, 2016; Erdogan & Webb, 2017). CAFs are thus involved in inducing epithelial–mesenchymal transition (EMT) and can promote the acquisition of stemness in breast cancer cells (Giannoni et al., 2010; Plaks, Kong & Werb, 2015). However, during the initial stages of formation of primary or metastatic lesions, the tumour microenvironment mostly contains normal fibroblasts (NFs). Some of these resting NFs can be activated due to inflammation, tissue damage, chronic fibrosis and other processes to become normal activated fibroblasts (NAFs) (Darby et al., 2016). However, very little is known regarding the function of the two populations of fibroblasts, CAFs and NAFs, during tumour formation and progression, especially regarding their effects on BCSCs.

As a preliminary investigation, we aimed to determine the roles of CAFs and NFs during the acquisition of BCSC phenotypes and to explore the similarities and differences between BCSCs that are induced by CAFs and NFs from a functional and observational perspective to lay the foundation for further mechanistic exploration.

Materials and Methods

Cells and cell culture

Cancer-associated fibroblasts and normal fibroblasts

For the primary culture of CAFs and NFs, tissues were collected from six breast cancer patients who underwent complete surgical resection of their tumours at Xuanwu Hospital, Capital Medical University. This study was approved by the appropriate Institutional Review Board and Human Ethics Committee, and all participants were adequately informed before the use of their samples. The donors’ clinical information is summarized in Table 1.

Table 1 Clinical information of the specimen donors.

Sample name	Patient’ age	Histological type	Largest diameter	ER (%)	PR (%)	HER-2	Ki67 (%)	Axillary lymph node metastasis	Cyclomastopathy	
CAF-g/NF-g	46	IDC	25 mm	+, 80	+, 40	Negative	15	Yes	Yes	
CAF-h/NF-h	33	IDC	33 mm	+, 95	+, 10	Negative	20	Yes	Yes	
CAF-i/NF-i	60	IDC	18 mm	+, 95	+, 80	Positive	5	No	Yes	
CAF-l/NF-l	44	IDC	26 mm	+, 80	–	Negative	25	Yes	Yes	
CAF-m/NF-m	47	IDC	30 mm	+, 95	+, 90	Negative	15	Yes	Yes	
CAF-n/NF-n	66	IDC	30 mm	+, 95	–	Negative	30	Yes	Yes	
Note:

Age represents the age of the donor at the time of surgery. The histological type of the tissue was invasive ductal carcinoma (IDC). Oestrogen and progesterone receptor status is indicated as ‘positive/negative’ with the corresponding percentage of positive cells. HER-2 positivity in the tissues was determined by fluorescence in situ hybridization (FISH).

Paired fibroblasts were isolated from the same patient from corresponding tumour and healthy breast tissues more than 3 cm away from carcinoma. After being harvested, tissues were stored in Dulbecco’s modified Eagle’s medium (DMEM) (Gibco, Waltham, MA, USA) supplemented with 1% penicillin-streptomycin (Gibco, Waltham, MA, USA) and immediately transported on ice to the laboratory. The tissues were mechanically chopped, washed three times with phosphate-buffered saline (PBS) (Gibco, Waltham, MA, USA) and enzymatically digested with prepared reagents for 9–15 h at 37 °C. The reagent mixture used for digesting NFs contained collagenase type I (Sigma, St. Louis, MO, USA) and hyaluronidase (Sigma, St. Louis, MO, USA), whereas that for digesting CAFs contained only collagenase. After the cell suspension was filtered with 100-mesh screens and centrifuged at 1,000 rpm for 4 min, the cell pellet was resuspended in the fresh DMEM containing 10% heat-inactivated foetal bovine serum (FBS) (BI). These cells were cultured at 37 °C in 5% CO2 and 95% air, and half of the medium was replaced three times a week. All CAFs and NFs were used in the experiments within eight passages.

Cell lines

The human breast cancer cell line MCF-7 was purchased from Cobioer Biosciences Corporation (Nanjing, China) and cultured in DMEM containing 10% FBS at 37 °C in a humidified atmosphere of 5% CO2 and 95% air.

Mammosphere culture and dissociation

In a published report (Shaw et al., 2012), MCF-7 cells had the highest mammosphere-forming efficiency (MFE) among all kinds of breast cancer cell lines. Thus, we used the mammosphere-forming capacity of cell lines derived from MCF-7 cells to purify and collect BCSCs. MCF-7 cells were cultured to 70–80% confluency and suspended in DMEM/F12 (Gibco, Waltham, MA, USA) containing 2% B27 (Gibco, Waltham, MA, USA), 20 ng/ml epidermal growth factor (EGF) (Gibco, Waltham, MA, USA) and 20 ng/ml recombinant basic fibroblast growth factor (bFGF) (PeproTech, Rocky Hill, NJ, USA) to form single cell suspensions. The cell suspensions were plated at a density of 2.5 × 105 cells/10 cm diameter dish with 10 ml culture medium. The cells were incubated in a humidified atmosphere at 37 °C and 5% CO2 for seven days without moving or disturbing the plates and without replenishing the medium. After a week, ‘primary mammospheres’ were obtained, which were then collected by gentle centrifugation (580 g, 4 min) and trypsinized with 0.05% trypsin/0.53 mM ethylenediaminetetraacetic acid (EDTA)-4Na (Gibco, Waltham, MA, USA). To ensure single cell suspension, the cells were passed three times through a 25 g needle on a syringe and inspected under a microscope. The ‘secondary mammospheres’ were plated at the same seeding density and incubated under the same conditions that were used for the generation of primary mammospheres.

Immunocytochemistry

Cancer-associated fibroblasts and NFs were cultured on coverslips at 50–60% confluency. Then, the fibroblasts were fixed in cold acetone for 7 min and incubated overnight with a primary antibody against α-SMA (1:100) (Abcam, Cambridge, UK) at 4 °C in a moist chamber. PBS was used as a control. After being washed three times with PBS, all coverslips were incubated with a peroxidase-conjugated secondary antibody for 15 min at room temperature, followed by incubation with 3,3′-diaminobenzidine. The coverslips with cells were then counterstained with haematoxylin for 2 min and placed on slides.

Immunohistochemical staining

Samples collected from the patients described above were fixed in formalin and embedded in paraffin. Then, the samples were cut into 4 μm sections with a microtome. All sections were air-dried overnight, dewaxed in xylene and dehydrated in alcohol before staining. The primary antibody against α-SMA was the same as that described above. Staining was performed according to the manufacturer’s instructions. Antigen retrieval was performed by pressure cooking the sections for 1 min in EDTA.

Preparation of conditioned medium

Cancer-associated fibroblasts and NFs were cultured to 70–80% confluency and were then washed with PBS. After that, cells were cultured in DMEM/F12 or DMEM with neither FBS nor other growth factors for 24 h. The two types of media were pipetted into a centrifuge tube and centrifuged at 1,000 rpm for 5 min, and the supernatant was collected and stored at −20 °C in small aliquots to avoid repeated freeze-thaw cycles. Before use, 10% FBS was added to the DMEM supernatant to form CM for the MCF-7 cell invasion assay. Similarly, EGF, bFGF and B27 were added to the DMEM/F12 supernatant to form CM for BCSC-related experiments.

Cell invasion assay

Invasion assays were conducted using 8 μm-pore Transwell inserts in 24-well plates (Corning). We added 25 μl of growth factor-reduced phenol red-free Matrigel (BD), which was diluted in a 1:4 ratio with DMEM, to Transwell inserts on ice and incubated the inserts at 37 °C without disturbance. After 2 h, 3 × 104 MCF-7 cells in 100 μl FBS-free DMEM were seeded in the upper chamber. The plates were divided into the following three groups: a group with CM-derived from CAFs with 10% FBS (CM-CAFs), a group with CM-derived from NFs with 10% FBS (CM-NFs) and a control group with DMEM and 10% FBS. After 48 h of incubation at 37 °C and 5% CO2, the cells remaining in the upper chambers were removed with a cotton swab, and the inserts were fixed in cold 4% formaldehyde for 5 min and stained with 5% crystal violet for 5 min. Then, the cells that had invaded to the lower surface of the membranes were counted from six random fields of views with an inverted microscope at 400× magnification. The assay was performed in triplicate.

Protein analysis

Sample preparation

Proteins in culture supernatants were purified using an acetone precipitation method. For whole-cell extractions, cell pellets were lysed with urea-containing lysis buffer (8 M urea and 100 mM Tris-HCl pH 8.0). Protease inhibitors (Pierce™; Thermo Fisher Scientific, Waltham, MA, USA) were added to protect proteins from degradation, and protein concentrations were measured using the Bradford assay (Eppendorf BioSpectrometer, Hamburg, Germany).

Trypsin digestion of proteins

For whole-cell proteome analyses, proteins were digested using the filter assisted sample preparation method as follows: the protein samples were incubated with 1 M dithiothreitol (DTT) at a final concentration of 5 mM for 30 min at 56 °C; then, iodoacetamide was added to the samples at a final concentration of 20 mM and incubated in the dark at room temperature. After incubation for half an hour, the samples were mixed with DTT at a final concentration of 5 mM and kept in the dark for another 15 min. Then, the protein samples were loaded onto 10-kD Microcon filtration devices (Millipore, Darmstadt, Germany) and centrifuged at 12,000g for 20 min and washed twice with urea-containing lysis buffer (8 M urea and 100 mM Tris-HCl pH 8.0) and twice with 50 mM NH4HCO3. Then, the samples were digested using trypsin at an enzyme to protein mass ratio of 1:25 overnight at 37 °C. Peptides were then extracted and dried (SpeedVac; Eppendorf, Hamburg, Germany).

Mass spectrometric analysis and data processing

Orbitrap Fusion liquid chromatography and tandem mass spectrometry (LC-MS/MS) analyses were performed on an Easy-n LC 1,000 LC system (Thermo Fisher Scientific, Waltham, MA, USA) coupled to an Orbitrap Fusion MS and a nano-electrospray ion source (Thermo Fisher Scientific, Waltham, MA, USA). Samples were dissolved in loading buffer (5% methanol and 0.1% formic acid) and loaded onto a 360 μm ID × 2 cm C18 trap column at a maximum pressure 280 bar with 12 μl solvent A (0.1% formic acid in water). Peptides were separated on a 150 μm ID × 10 cm C18 column (1.9 μm, 120 Å; Dr. Maisch GmbH, Ammerbuch-Entringen, Germany) with a series of adjusted linear gradients according to the hydrophobicity of fractions with a flow rate of 500 nl/min. MS analysis was performed in a data-dependent manner with full scans (m/z 300–1,400) acquired using an Orbitrap Mass Analyser at a mass resolution of 120,000 at an m/z of 200. The top data-dependent speed mode was selected for fragmentation in the human collecting duct cell at normalized collision energy of 32%, and then fragment ions were transferred into the ion trap analyser with an automatic gain control target of 5 × 103 counts and maximum injection time of 35 ms. The dynamic exclusion of previously acquired precursor ions was enabled at 18 s.

Raw data were analysed in the Firmiana data analysis environment for protein identification and label-free quantification using a target-decoy approach to identify peptides and proteins at an false discovery rate (FDR) <1% against the human protein RefSeq database (released 2013.07.01). Acetylation (protein N terminus) and oxidation (methionine), carbamidomethylation (cysteine) were set as variable modifications.

Sphere formation assay

Fourth and higher generation mammospheres were used for the experiments. Cell suspensions prepared as described above in CM and plated at a density of 500 cells/cm2 in ultra-low attachment six-well plates (2 ml of mammosphere medium per well). The plates were divided into three groups based on the medium used, namely, CM-CAFs, CM-NFs and DMEM/F12-control medium. All DMEM/F12-control medium used in mammosphere experiments contains DMEM/F12, EGF, bFGF and B27. After six days, we counted the number of mammospheres that were more than 50 μm in diameter and calculated MFE based on the following formula: MFE (%) = (number of mammospheres per well/number of cells seeded per well) × 100. The assay was performed in triplicate.

Flow cytometry

CD24, CD44 and ALDH1 expression was analysed in cells harvested from fourth and higher generation mammospheres dissociated after six days of culture in CM. After preparing single cell suspensions in PBS, 1 × 106 cells were counted and resuspended in staining buffer and incubated for 30 min with anti-CD44 and anti-CD24 antibodies (BD) at the manufacturer’s suggested dilutions at 4 °C in the dark. The labelled cells were washed and resuspended in PBS on ice. Additionally, 1 × 105 cells from the samples were subjected to the ALDEFLUOR assay (Millipore, Darmstadt, Germany). All samples were analysed with a flow cytometer (BD, San Jose, CA, USA). Cells incubated with isotype control antibodies corresponding to each fluorescent probe were used as negative controls. In addition, cells incubated with diethylaminobenzaldehyde, an inhibitor of ALDH1, were used as negative controls for the ALDEFLUOR assay. The assay was performed in triplicate.

Immunofluorescence staining

For immunofluorescence, mammospheres were cultured on coverslips in CM for six days, fixed in cold acetone for 7 min and then stored in PBS. After being washed, the mammospheres were blocked in methanol containing hydrogen peroxide for 10 min. Then, the mammospheres were incubated with primary antibodies (ALDH1A1; Proteintech, Rosemont, IL, USA) (CD44; Cell Signaling, Danvers, MA, USA) at 4 °C overnight in a humid chamber. After being washed three times with PBS, the cells were incubated for 1 h with secondary antibodies. The coverslips were mounted with mounting medium containing 4’,6-diamidinophenyl-indole (DAPI). Fluorescence signals were observed and images were obtained with a fluorescence microscope. The assay was performed in triplicate.

Statistical analysis

Data were presented as the means ± SD for at least three independent experiments for every group. Statistical analyses were performed with SPSS 16.0 software. Statistical differences were analysed using two-tailed Student’s t-tests. Differences were considered significant at P < 0.05.

Results

Cancer-associated fibroblasts and their paired NFs were successfully isolated from six primary breast cancer samples and paired normal breast tissue specimens, respectively. The primary fibroblasts were successfully cultured in DMEM containing 10% FBS, and these fibroblasts could grow well for at least nine passages. Both CAFs and their paired NFs had similar spindle-like morphology (Figs. 1A and 1B). Depending on the age of the patients from whom the samples were obtained, senescent cells were observed in different passages, especially after the sixth passage. The cultured CAFs/NFs and breast cancer/adjacent normal tissues all expressed α-SMA, which is considered myofibroblast-specific marker (Figs. 1C–1F). Then, a culture system with CM was established to identify the different roles of CAFs and NFs in the tumour’s paracrine microenvironment. In contrast with the cells grown in culture systems with CM-NFs and DMEM with 10% FBS, MCF-7 cells grown in culture systems with CM-CAFs displayed increased invasion (Figs. 1G–1I). Therefore, CAFs and NFs could be isolated from different tissues and exerted different functions in the tumour microenvironment.

Figure 1 Cell morphology and α-SMA expression cannot be used to distinguish CAFs and NFs, but these fibroblasts differ in their biological functions.

(A and B) CAFs and NFs are both morphologically characterized as large spindle-shaped cells with indented nuclei. Bar: 100 μm. (C and D) Immunohistochemical staining for α-SMA in CAFs and NFs. Both samples are positive for α-SMA expression. Bar: 100 μm. (E and F) Immunohistochemical staining for α-SMA in a cancer tissue and a paracancerous tissue more than 3 cm away from carcinoma. Bar: 100 μm. (G–I) Invasion of MCF-7 cells was affected by CM. Bar: 100 μm.

We collected samples of whole-cell extracts from six paired primary fibroblast lines cultured at 90–100% confluence. Label-free LC-MS/MS quantification was used to characterize the differences in protein expression between CAFs and NFs. In total, 7,208 proteins were identified. With the criteria of up-regulation in at least four paired experiments and a fold-change >2,200 proteins were up-regulated in CAFs, and 5,037 proteins were up-regulated in NFs. Quantitative data for the proteins up-regulated in CAFs is represented in the form of a heat map (Fig. 2A). The functions, biological processes (Fig. 2B) and Kyoto Encyclopaedia of Genes and Genomes (KEGG) pathways (Fig. 2C) associated with these proteins were also analysed. The proteomic data illustrated differences in protein expression between paired CAFs and NFs.

Figure 2 CAFs and NFs are heterogeneous in terms of their protein expression profiles, cellular functions and related signalling pathways.

(A) Heterogeneity in the expression of selected proteins (fold-change < −2 or >2, P < 0.05) between CAFs and NFs. Blue represents down-regulated expression, and red indicates up-regulated expression. All data are transformed by log2 operation. (B) Gene ontology (GO)-based enrichment analysis of up-regulated proteins in CAFs (orange bars) and in paired NFs (blue bars), including cell component (CC), biological process (BP) and molecular function (MF). (C) KEGG pathway analysis for identifying major biological pathways that were different between CAFs (orange bars) and NFs (blue bars).

To explore the role of CAFs and NFs in BCSCs, we employed a culture system with CM from corresponding fibroblasts and analysed the effect of the CM on mammosphere formation by MCF-7 cells. Single cell suspensions of MCF-7 cells were cultured independently in CM-CAFs, CM-NFs and DMEM/F12-control medium as mentioned above, without disturbing the plates. Mammospheres, with rounded shape and smooth boundaries, were observed by microscopy after six days of culture (Fig. 3A). Mammospheres greater than 50 μm in diameter were counted and included in statistical analyses. The average size of the mammospheres was not noticeably different among the groups. However, the number of mammospheres formed in the groups with CM-CAFs and CM-NFs was approximately 3.84-fold and 2.85-fold higher, respectively, than that in the control group (Fig. 3B). Cells cultured in CM-CAFs had a greater MFE than the cells cultured in CM-NFs (P < 0.05) (Fig. 3C). Thus, CM-CAFs demonstrated greater capability of inducing mammosphere formation than did CM-NFs.

Figure 3 Mammosphere formation by MCF-7 cells cultured in CM.

(A) Mammosphere formation in MCF-7 cells in the control, CM-NFs (cultured in CM-NFs) and CM-CAFs (cultured in CM-CAFs) groups. Bar: 100 μm. (B) Number of mammospheres greater than 50 μm in diameter in each group. (C) MFE in each group.

We performed serial flow cytometry to explore the relationship between CD44+CD24− and ALDH1+ cell populations, corresponding to the expression of markers that are widely used to characterize BCSCs. Overall, ALDH1+ BCSCs were highly enriched in the CM-CAFs group (Figs. 4A–4C), whereas cells in the CM-NFs group were enriched for the CD44+CD24− population (Figs. 4D–4F). Based on the ALDEFLUOR assay, the proportion of ALDH1+ cells increased in CM-CAFs relative to those in the groups with both DMEM/F12-control medium and NF-conditioned medium (P < 0.01).

Figure 4 CM from CAFs and NFs can induce phenotypic transition in BCSCs.

(A–C) ALDH1 expression in MCF-7 mammospheres from the three groups based on the ALDEFLUOR assay followed by fluorescence activated cell-sorting (FACS). (D–F) CD24 and CD44 expression in MCF-7 mammospheres from the three groups as detected by FACS. (G and H) Localization of CD44 (green), ALDH1 (red) and DAPI (blue) in mammospheres adhered to glass slides as assessed by immunofluorescence. Bar: 100 μm.

However, there was no significant difference in the proportion of the CD44+CD24− population between cells cultured CM-CAFs and cells cultured on DMEM/F12-control medium (P = 0.181), but the proportion of CD44+CD24− BCSCs was markedly increased among the cells cultured in CM-NFs (P < 0.05).

Further, the expression of CD44 and ALDH1 was examined using immunofluorescence in the two groups of mammospheres cultured in different CM. The expression of ALDH1 in the cytoplasm was enhanced by paracrine stimulation via CM-CAFs (Fig. 4G). Additionally, the mammospheres displayed significantly enhanced CD44 expression when cultured in CM-NFs (Fig. 4H).

Discussion

In the present study, we used proteomics methods to investigate the protein profiling of breast cancer CAFs and their normal counterpart NFs, which were isolated from breast cancer and paired with paracancerous tissues. We found that the differences in expression patterns between CAFs and NFs were mainly focused on the regulation of cellular extracellular matrix remodelling, metabolism and regulation of TGF-β pathways. We also found that the CM from CAFs and NFs has different biological functions in promoting the invasiveness of breast cancer cells, which is consistent with previous reports (Chen et al., 2012). Interestingly, our work demonstrated that the CM from both CAFs and NFs could increase the MFE of MCF-7 cells. Compared with the CM from CAFs, the CM from NFs could induce an increase in the proportion of CD44+CD24− BCSCs. Considering that the evidence-based data, which illustrates the effect of fibroblasts on the phenotypic transformation of BCSCs to various states, are limited and tend to rely on studies focusing on CAFs (Gunaydin, Dolen & Kesikli, 2013; Lisanti et al., 2014), the present study might broaden our horizons of the critical roles of both CAFs and NFs in the dedifferentiation and phenotype conversion of breast cancer cells.

Though several studies have already reported several classic biomarkers for CAFs, few of them are proved to be specific, such as α-SMA (Qiao et al., 2016). In this study, immunohistochemical staining was applied to detect the expression levels of α-SMA in paraffin-embedded breast cancer tissues and para-carcinoma tissues and in isolated CAFs/NFs, and we found that both CAFs and NFs expressed α-SMA, as observed in previous studies (Huang et al., 2010), despite some differences in staining patterns. This finding drives us to conduct a systemic evaluation of protein expression between CAFs and NFs. Heat map cluster analysis demonstrated the differences between the protein expression profiles of these two fibroblast types, which were consistent with previously reported GeneChip data (Al-Rakan et al., 2013; Pasanen et al., 2016). Furthermore, Gene ontology analysis revealed that compared to the proteins up-regulated in NFs, the proteins up-regulated in CAFs were associated with cytoskeleton formation, cell adhesion and integrin-mediated signalling pathways. Moreover, the identified proteins regulate the TGF-β signal pathway. However, the proteins down-regulated in CAFs were associated with gene transcription and translation, lysosome transport, oxidative stress, mitochondria and cell morphology, among other processes. KEGG analysis showed that the pathways up-regulated in CAFs were associated with intercellular adhesion and connection. These pathways can also regulate the formation of the cytoskeleton. The pathways down-regulated in CAFs were associated with RNA degradation, RNA transport and fatty acid metabolism. Our data provide a comprehensive view of the protein profiling of CAFs and NFs, which might provide valuable biomarkers for CAFs. In fact, data mining and validation to identify intracellular and secreted extracellular protein expression profiles in CAFs and NFs are ongoing.

There is a consensus that CAFs can affect the invasive behaviour of breast cancer cells (Luo et al., 2015). However, the underlying mechanism of how CAFs affect the invasive potential of breast cancers is still uncertain. In the present study, we proved that the CM from CAFs could promote the invasiveness of the breast cancer cell line MCF-7, which could be explained by our proteomic results that the well-known TGF-β oncogenic signalling pathway was significantly up-regulated in CAFs. We hypothesized that CAFs could enhance the metastatic potential of breast cancer cells though the paracrine stimulation of TGF-β. In fact, there have been reports of TGF-β supported tumour invasion by several mechanisms, for example, stimulating the trans-differentiation of epithelial cancer cells into migratory mesenchymal cells, namely the EMT process (Yu et al., 2014). In addition, in some human breast cancer cells, organization of the invadopodia or degradation of the extracellular matrix (ECM) requires extracellular or intracellular signal-regulated kinase signalling through phosphatidylinositide 3-kinase and Src kinase, which is also governed by TGF-β (Mandal, Johnson & Wheelock, 2008; Pignatelli et al., 2012). In our subsequent work, we will investigate whether TGF-β signalling contributes to CAF-induced acquisition of invasion potential in breast cancer cells. Moreover, details of the mechanism will be illustrated.

The highlight in our study is that we clarified the roles of CAFs and NFs in the induction of cancer stem cells. Compared with the DMEM/F12-control medium, the CM from CAFs and NFs could significantly enhance the MFE of MCF-7 cells. Among the CM from the two kinds of fibroblasts, the mammosphere-promoting effect was stronger for the CM-CAFs. These data are also consistent with the findings of stem cell-related studies in which fibroblasts are used as feeder cells or in co-culture systems cells to maintain the characteristics of stem cells and to induce the dedifferentiation of non-stem cells (Chen et al., 2014; Zhang et al., 2011). Further, using flow cytometry and immunofluorescence, we found that the cytokines or exosomes secreted by CAFs and NFs could induce the phenotypic transformation of BCSCs. ALDH1, CD44 and CD24 are common markers used for the identification of BCSCs. Previous studies on the effects of tumour stroma on BCSCs did not investigate the effect on phenotypic transformation. Liu et al. proposed that although ALDH1+ BCSCs and CD44+CD24− BCSC are overlapping cell populations, they are not identical (Liu et al., 2014). The former cell population is highly proliferative, while a majority of CD44+CD24− BCSCs are typically quiescent. A rare proportion of ALDH1+ and CD44+CD24− BCSCs displays very strong tumourigenicity in vivo (Croker et al., 2009). Immunofluorescence assay showed that ALDH1+ BCSCs and CD44+CD24− BCSCs are localized in different regions within the tumour tissue. The former cell population is typically located in the centre of the tumour, while the latter is located more at the periphery. In this study, the proportion of ALDH1+ BCSCs was relatively increased in response to CM-CAFs; however, BCSCs treated with CM-NFs were more likely to undergo phenotypic transformation into CD44+ populations. Thus, we hypothesized that the effect of fibroblasts on BCSCs may be multidirectional. Fibroblasts at different states may have different effects on BCSC populations and thus promote diverse biological processes. As shown in this study, CM-CAFs could promote the invasion of tumour cells in a paracrine manner, while cytokines secreted by NFs suppressed local invasion of tumour cells. Since accumulating evidence has demonstrated a crucial role of TGF-β in the regulation of cancer stem cell function (Watabe & Miyazono, 2009), it has been observed that TGF-β can affect the generation and outcome of normal and malignant stem cells through various mechanisms (Watabe & Miyazono, 2009). In breast cancer, previous studies have reported that the activation of the TGF-β pathway can enrich CD44+ BCSCs (Shipitsin et al., 2007). However, other groups have also found that active TGF-β signalling can increase a subpopulation of stemness cells that express ALDH1 (Zheng et al., 2014), suggesting that the regulatory function of TGF-β in BCSCs is complicated. In our present work, we found that TGF-β activated fibroblasts, namely the CAFs, could increase the proportion of ALDH1+ BCSCs via their conditioned medium. These results hinted that TGF-β as a paracrine factor might be effective as a regulator in the transformation of BCSC subpopulations from CD44+ to ALDH1+. This study expanded on the conclusion that fibroblasts can promote the generation of cancer stem cells. Further, we will study whether such a subpopulation conversion is associated with tumour progression, such as tumour invasion, and the underlying mechanism of this process.

This study has some potential limitations. First, although in vivo experiments are the gold standard for verifying the self-renewal ability of cancer stem cells, we cannot prevent phenotypic changes induced by tumour cells in NFs in vivo. We do not have data from in vivo experiments investigating the effects of NFs and CAFs on cancer stem cells. In addition, this research only proves that CM from CAFs or NFs enrich the BCSCs, however future study is needed to find out the exact approach that induces this promoting effect. It may the result of increased proliferation of BCSCs, represent of low apoptosis ration, or the conversion of non-stem cells into stem cells. At last, because this study is preliminary, the results merely describe the effect of CAFs/NFs on the phenotype of BCSCs from a functional perspective. Further molecular analyses are needed to explore the associated cytokines or pathways, and we will pursue this direction in the future.

Conclusion

In summary, our preliminary study validated the existence of heterogeneity among CAFs and NFs from a proteomics perspective. Both types of fibroblasts promoted the self-renewal capacity of BCSCs and induced the phenotypic transformation of non-stem cells to cancer stem cells. However, during this process, CAFs and NFs showed different extents of promotive ability and different directions of phenotypic transformation in BCSCs. Further investigation on the effect of NFs on cancer stem cells may provide a new direction for the elucidation of the initial stages of tumour metastasis. Additionally, insights into the effects of CAFs/NFs on BCSCs may provide new directions in the study of breast cancer treatment. Based on the functional evidence obtained in this study, further studies will be carried out to explore the molecular mechanisms and relative pathways involved in the phenotypic transformation of BCSCs.

Supplemental Information

Supplemental Information 1 Raw data of cell culture.

Click here for additional data file.

Supplemental Information 2 Raw data of FCS.

Click here for additional data file.

Supplemental Information 3 Raw data of ICC.

Click here for additional data file.

Supplemental Information 4 Raw data of IF.

Click here for additional data file.

Supplemental Information 5 Raw data of IHC.

Click here for additional data file.

Supplemental Information 6 Raw data of invasion assay.

Click here for additional data file.

Supplemental Information 7 Raw data of mammosphere formation assay.

Click here for additional data file.

Supplemental Information 8 Raw data of protein analysis.

Click here for additional data file.

We thank Prof. Jun Qin, Ph.D., State Key Laboratory of Proteomics, National Center for Protein Sciences, Beijing Proteome Research Center, Yeqing Cui, medical laboratory technician of Xuanwu Hospital Surgery Lab, Jiaying Yu, Ph.D., Institute of Biophysics, Chinese Academy of Sciences, for their expertise and help during this project, and Dr. Tian Lan, Dr. Ye Cheng and Dr. Fei Gao for proofreading and language check.

Additional Information and Declarations

Competing Interests

Author Contributions

Human Ethics

Data Availability

The authors declare that they have no competing interests.

Bixiao Wang conceived and designed the experiments, performed the experiments, analyzed the data, contributed reagents/materials/analysis tools, prepared figures and/or tables, authored or reviewed drafts of the paper, approved the final draft.

Chunfang Xi performed the experiments, analysed the data, contributed reagents/materials/analysis tools, authored or reviewed drafts of the paper.

Mingwei Liu performed the experiments, contributed reagents/materials/analysis tools.

Haichen Sun performed the experiments.

Shuang Liu performed the experiments.

Lei Song analysed the data.

Hua Kang conceived and designed the experiments, authored or reviewed drafts of the paper, approved the final draft, guide the experiment.

The following information was supplied relating to ethical approvals (i.e., approving body and any reference numbers):

The study was approved by the Institutional Review Board and Human Ethics Committee of Xuanwu Hospital, Capital Medical University.

The following information was supplied regarding data availability:

The raw data are provided in the Supplemental Files.

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
