# Peer review of "Breast fibroblasts in both cancer and normal tissues induce phenotypic transformation of breast cancer stem cells: a preliminary study"

_PeerJ, doi:10.7717/peerj.4805_

## Round 0.1 · original submission · Major Revisions

The manuscript addressed the role of breast fibroblasts in tumor and non-tumor in phenotypic transformation of breast stem cells. The subject is interesting and worth of investigation, however the number of samples analyzed (n=6) is too small to support any conclusions, especially if we consider the heterogeneity of the tumors. For example, only one sample is HER2 positive, and PR expression is also highly variable among samples.

The experimental design need a better description.
The proteomic characterization of CAFs and FNs also need to be improved, but this characterization is not new in the literature.

I believe the authors should carefully review their manuscript considering both reviewers comments.

Reviewer 1 ·

Basic reporting

Comments
The manuscript by Bixiao wang et.al aimed to address the role and effect of NFs and CAFs in the phenotypic transformation of BCSCs. There are some technical concerns which limit the originality of the data presented.
 The authors mentioned about the use of CD44+CD24- BCSCs and association of BRCA1 mutational status. There are several controversies regarding this. It has already been reported by Somasundaram et al., 2016 that the best stem cell marker for BRCA1 mutated cancer would be ALDH1 and not CD44+CD24-
 The authors failed to put the reference appropriately in the Materials and Methods (line 141) for mammosphere culture and dissociation.
 Moreover, the histological type of the tissue was shown infiltrating ductal carcinoma initially and later on in discussion part the authors mentioned that the samples were collected from breast cancer patients of mammary gland hyperplasia which actually lacks the clarity about the type of tumor analyzed and creates confusion.

Experimental design

There is a procedural problem in preparation of Condition Media (CM) as the authors are adding 10% FBS to the CM before use which actually dilutes the properties of CM and the addition of growth factors seem to be unnecessary.

The methods described were not listed properly and they did not to specify which is the control medium used.

Table-1 regarding the clinical information described in Materials and Methods shows the uneven hormonal status of the samples used for the study. All the samples should be of same kind eg. either triple negative or triple positive etc. This is because only a few samples are analyzed in the study.

Validity of the findings

In the preliminary experiment, to distinguish between CAFs and NFs, the authors have analysed the expression status of αSMA. Since αSMA is not a conclusive marker to differentiate both, more marker genes should be included and analysed for a better confirmation. Also the IHC of αSMA shows non-specificity which could be a procedural problem (Figure -1).

The validation of heterogeneity among CAFs and NFs has already done and reported, so further validation of the same in the study is not relevant in both results and discussion.
Discussion is too long. It can be summarized with more clarity and should be precise.

Comments for the author

As indicated above

Reviewer 2 ·

Basic reporting

The manuscript describes interesting approach in the study of the role of CAFs and NFs in the breast cancer stem cells phenotype. Article structure, language and figures are of good level.
According to the authors, the experiments aimed “to explore the similarities and differences between BCSCs that are induced by CAFs and NFs from a functional and observational perspective to lay the foundation for further mechanistic exploration”. The main questionable points are:
-What is the direct evidence of BCSCs induction by CAFs and NFs?
- The effects of conditioned medium from CAFs and FNs on MCF-7 showed increased number of mammospheres compared with controls. What is the meaning of a high number of mammospheres? Since the initial cell number is the same, did CM induce cell proliferation or did inhibit cell dead?

Experimental design

They used cultures of cells from 6 breast cancer patients and only in the protein expression analysis they compared the CAFs and FNs from all six. KEGG pathway analysis identified the major biological pathways in the 2 types of cells, showing that they were different in spite of both being positive for α-SMA. It would be very interesting to discuss in deep the meaning of protein expression in relation to cell phenotypes. For example, why the pathways up-regulated in CAFs were associated with intercellular adhesion and connection? Are these pathways associated with the formation of the cytoskeleton only in CAFs?
Minor observation:
Fig 1B- Are they both CAFs and NFs in same magnification?
Line 402- “Fibroblasts are the predominant cell type in the extracellular matrix.” Please correct to: Fibroblasts are the predominant cell type in the connective tissue or in stroma

Validity of the findings

no comment

---

## Round 0.2 · Minor Revisions

Please address the remaining comments raised by reviewer 2.

Reviewer 1 ·

Basic reporting

The Authors have incorporated the recommendations by the reviewer in the revised manuscript.
Therefore the manuscript may be accepted in the journal Peer J.

Experimental design

I appreciate for the authors for properly designing the experiments.

Validity of the findings

Revised manuscript has all the recommendations incorporated in their finding.

Comments for the author

The manuscript may be accepted in the journal Peer J.

Reviewer 2 ·

Basic reporting

The revised version of manuscript is improved by the inclusion of some missed informations, mainly in M&M, and by the discussion reformulation, as well. Detailed comments on the reviewer’s questions are in the rebuttal letter, however some of them are are not clear enough. In replying to my questions, they claimed that “….the number of mammospheres in the groups with CM-CAFs and CM-NFs were obviously higher”. Sorry, these data are not a direct evidence of BCSC number. The efficiency of mammosphere formation is a complex index resulting from several factors. For example, an increasing in mammosphere formation may represent low apoptosis ration, increased proliferation of early progenitor cells or stem cells, or an increase in stem cell self-renewal.
Absolutely dispensable the inclusion of Figure 1S.
Please, check the English .

Experimental design

no comment to add

Validity of the findings

no comment to add

---

## Round 0.3 · accepted · Accept

The revised manuscript addressed the comments raised by the referee and is now suitable for publication in PeerJ.